# Diagnosis, clinical features, and mortality risk factors in a Chinese cohort with pulmonary mucormycosis

**Junjun Wan**[1], **Teng Liu**[2], **Fang Li**[3], **Shaohua Xu**[1]*

1 Department of Respiratory Medicine, the Second Hospital of Shandong University, Jinan, China,
2 Department of Ultrasound, Qilu Hospital of Shandong University, Jinan, China, 3 Department of Cardiac Surgery Intensive Care Unit (ICU), Shandong Provincial Hospital Aliated to Shandong First Medical University, Jinan, China

* xush88880@163.com

## Abstract

### Background

Pulmonary mucormycosis is a rare and often fatal fungal infection. Identifying high-risk factors for pulmonary mucormycosis holds the potential to improve patient outcomes. This study aimed to identify the clinical characteristics and risk factors associated with pulmonary mucormycosis outcomes in a Chinese cohort.

### Materials and methods

A retrospective analysis was conducted on 37 patients diagnosed with pulmonary mucormycosis, focusing on clinical records, laboratory findings, and computed tomography (CT) imaging. Diagnosis was primarily based on histopathology or next-generation sequencing.

### Results

The median age of the patients was 55 years, and the most common underlying conditions were hematological malignancies, diabetes, and organ transplantation. Imaging frequently revealed bilateral lung involvement with ground-glass opacities and nodular lesions. The overall mortality rate was 29.7%, with significant risk factors for 90-day mortality including hypertension (Hazard Ratio [HR] = 3.36, 95% Confidence Interval [CI] = 1.01–11.12, P = 0.048), organ transplantation (HR = 4.93, 95% CI = 1.48–16.4, P = 0.009), and immunosuppression (HR = 8.83, 95% CI = 1.13–69.14, P = 0.038).

### Conclusions

Early suspicion and timely diagnostic measures, such as biopsy or metagenomic sequencing, are crucial for improving patient outcomes. These findings underscore the importance of recognizing and managing pulmonary mucormycosis in high-risk populations.

**Data availability statement:** The data supporting this study cannot be made publicly available due to legal and ethical restrictions. However, researchers may request access to the data by contacting the Ethics Committee of the Second Hospital of Shandong University (Approval No. KYLL2024LW035) for approval.

Data access requests should be sent to:
18366117163@163.com.

**Funding:** The author(s) received no specific funding for this work.

**Competing interests:** The authors have declared that no competing interests exist.

# 1 Introduction

Mucormycosis, often referred to as zygomycosis, is an acute or subacute invasive fungal infection caused by members of the Mucorales order [1,2]. These fungi are ubiquitous in soil, decaying organic matter, and animal excrement environments. While generally non-pathogenic to healthy individuals, Mucorales can become opportunistic pathogens, particularly in immunocompromised individuals, such as those with diabetes mellitus, organ transplantation, or neutropenia [3]. The primary route of infection is through the inhalation of sporangiospores, leading to pulmonary infection.

The incidence of the deadly mucormycosis has surged during the Coronavirus Disease 2019 (COVID-19) pandemic, elevating public health concerns and increasing clinical awareness. However, diagnosing mucormycosis remains challenging due to its rarity and the overlap of its symptoms with other conditions, often leading to delays in diagnosis and treatment. Mucormycosis can manifest in various forms, including rhino-orbit-cerebral, pulmonary, cutaneous, gastrointestinal, and disseminated types [4]. This diagnostic complexity contributes to the high mortality rate [5] with the 90-day mortality rate for patients with pulmonary mucormycosis ranging from 45% to 57% [6–9].

Most patients present with prolonged fever, though some may remain asymptomatic [10]. On chest CT, pulmonary mucormycosis may appear as ground-glass opacity, halo sign, crescent sign, thick-walled cavities, or fungal balls [11]. Early diagnosis is crucial for improving outcomes, however, current diagnostic methods' low sensitivity and specificity complicate early detection [12].

In this retrospective single-center study, we review the experience of the Second Hospital of Shandong University in treating patients with pulmonary mucormycosis. The primary objective of the study was to investigate the clinical manifestations, radiological features, treatment strategies, and outcomes of patients with pulmonary mucormycosis, with a particular focus on identifying factors that influence clinical outcomes. These cases highlight the critical importance of prompt diagnosis and the implementation of combined therapy in managing this life-threatening infection.

## Materials and methods

### Ethical considerations

This study was conducted following the ethical principles outlined in the Helsinki Declaration and was approved by the Research Ethics Committee of the Second Hospital of Shandong University (No. KYLL2024LW035). Due to the study's retrospective nature and the absence of identifying information, written informed consent was waived.

### Patients

This study included all cases of pulmonary mucormycosis diagnosed at our institution between 2018 and 2023. Data were retrospectively extracted from electronic medical records, covering demographic characteristics, infection sites, host factors, underlying diseases, diagnosis, clinical course, management, and outcomes. Clinical outcomes were defined as either survival or death within 90 days after diagnosis.

Data were accessed for research purposes between July 1, 2024, and July 15, 2024, and all information was handled in accordance with institutional guidelines for data privacy and security.

## Criteria for pulmonary mucormycosis

We applied the criteria set forth by the European Organization for Research and Treatment of Cancer and Mycoses Study Group (EORTC/MSG) to classify cases of invasive fungal disease as proven, probable, or possible [3]. The study included all consecutive subjects aged 18 years or older with proven or probable pulmonary mucormycosis. Proven pulmonary mucormycosis was identified in cases where there was both clinical and radiological suspicion of invasive mold infection, supported by the observation of aseptate hyphae or the confirmed growth of Mucorales from typically sterile sites, such as lung tissue or pleural fluid. Probable pulmonary mucormycosis was diagnosed in patients who exhibited consistent clinical and imaging findings, along with the detection of aseptate hyphae or positive Mucorales cultures from sputum or bronchoalveolar lavage fluid [13]. Patients who presented with host factors and clinical characteristics indicative of mucormycosis but lacked fungal evidence were classified as having "possible" fungal infections; these patients were excluded from the analysis due to the lower diagnostic certainty associated with this classification.

## Laboratory studies

Laboratory evaluations included complete blood count (CBC), C-reactive protein (CRP), procalcitonin (PCT), albumin (ALB), and the galactomannan (GM) test. All 37 patients were diagnosed through a combination of microbial culture, metagenomic sequencing, and/or histopathological analysis. Most patients underwent bronchoscopy with bronchoalveolar lavage (BAL), and clinical samples (including sputum and BAL fluid) were cultured under both aerobic and anaerobic conditions. Fungal cultures were performed on Sabouraud dextrose agar (SDA) medium, incubated at 30°C. Once colonies formed, macroscopic identification of colony morphology was performed, followed by microscopic examination using lactophenol cotton blue staining for species identification.

NGS testing (Next-Generation Sequencing) is a high-throughput and highly efficient DNA or RNA sequencing technology capable of parallel sequencing multiple DNA fragments in a short period of time. The DNA from the alveolar lavage fluid was extracted and processed for amplification and sequencing. The sequencing data were then assembled, annotated, and classified to determine the microbial species present in the samples.

Histological sections obtained from bronchoscopic or surgical biopsies were stained with periodic acid-Schiff (PAS) or hexamine silver, which enhanced the visualization of fungal elements. In cases of mucorales infection, characteristic hyphae were observed in the tissue samples.

## Radiological assessment

Chest CT scans were performed for all patients, however, cranial MR examinations were conducted only for patients presenting with head-related symptoms. The images were independently reviewed by two experienced radiologists. For patients diagnosed with pulmonary mucormycosis, the following CT findings were recorded: nodules, masses, cavities, consolidations, ground-glass opacities, reverse halo signs, lymphadenopathy, pleural effusions, and the distribution of lesions within the lungs.

## Treatment strategies and clinical outcomes

The treatment of mucormycosis includes surgical intervention, antifungal therapy, and management of underlying diseases. Whenever feasible, we perform surgical interventions on the lesions, including local debridement or resection of infected tissues or organs. Systemic antifungal therapy is essential, with treatment regimens including liposomal amphotericin B (LAmB, 3–6 mg/kg daily), amphotericin B deoxycholate (AmB, 0.5–1 mg/kg daily), isavuconazonium (200 mg

three times daily on days 1–2, followed by 200 mg once daily from day 3 onwards), and posaconazole oral suspension (400 mg twice daily).

Simultaneous management of underlying conditions is critical for improving treatment outcomes. In patients with underlying diseases, we implemented corresponding management measures. This involves optimizing blood glucose control, boosting granulocyte counts, and reducing or discontinuing glucocorticoids or immunosuppressive drugs where possible. The duration of antifungal therapy is individualized, based on the patient's clinical response and lesion size.

Clinical outcomes were evaluated 90 days after the diagnosis of mucormycosis and categorized as survival or death.

### Statistical analysis

All statistical analyses were performed using R Statistical Software (Version 4.2.2, http://www.R-project.org, The R Foundation) and the Free Statistics Analysis Platform (Version 1.8, Beijing, China). Categorical variables were compared using Fisher's exact test, and continuous variables were analyzed using the Wilcoxon rank-sum test. The univariable Cox regression model was utilized to compare the characteristics between surviving and deceased patients. Survival curves for patients with different immune statuses were generated using the Kaplan-Meier method. A two-tailed P-value of < 0.05 was considered statistically significant.

## Results

### Demographic characteristics and clinical manifestations

A total of 37 patients were included in the analysis, with 9 patients (24.3%) classified as having proven pulmonary mucormycosis and 28 patients (75.7%) classified as having probable pulmonary mucormycosis, according to the EORTC/MSG definitions for invasive fungal diseases (Table 1). The demographic characteristics, underlying conditions, predisposing factors, treatment regimens, and outcomes of these patients are summarized in Table 2.

The median age of the patients was 55.2 ± 16.2 years (22–84 years), and 67.6% of the cohort were male. Among the 37 patients with pulmonary mucormycosis, two also had concomitant rhino-orbital mucormycosis. The most common underlying condition was diabetes (n = 14, 37.8%), followed by hematological disorders (n = 11, 29.7%) and solid organ transplantation (n = 7, 18.9%). Of the seven patients who had undergone organ transplantation, five had received a renal transplant, and two had received a bone marrow transplant. Additionally, four patients (10.8%) had neutropenia. Up to 81% of patients with pulmonary mucormycosis had concurrent infections, including 3 with COVID-19, 1 with influenza A, 2 with rhinovirus, and 1 with Mycobacterium tuberculosis. Additionally, 9 patients had bacterial infections, 6 had aspergillosis, 3 had both viral and bacterial infections, and 5 had co-infections of bacteria and Aspergillus.

Clinical outcomes were assessed at 90 days post-diagnosis, with a 90-day mortality rate of 33.6% (11/37). No cases were lost to follow-up at the 90-day mark. Patients who had undergone organ transplantation appeared to have a significantly shorter survival time compared to those who had not received a transplant(P < 0.05). The clinical characteristics of pulmonary mucormycosis patients with different outcomes are detailed in Table 2.

The clinical manifestations of pulmonary mucormycosis were nonspecific. The most common symptoms included fever (70.3%) and cough (45.9%), followed by chest tightness (35.1%) and hemoptysis (29.7%). The median time from hospital admission to diagnosis was 13 days. Of the 37 patients, 31 (83.8%) had an acute course of disease (≤1 month), four (10.8%) had a subacute course (1–3 months), and two (5.4%) had a chronic course (>3 months). There were no significant differences in clinical symptoms or diagnostic levels between survivors and non-survivors (P > 0.05).

### Computed tomographic presentation

The CT findings in patients with pulmonary mucormycosis are varied, including nodules, masses, cavities, consolidations, ground-glass opacities, reverse halo signs, and lymphadenopathy, as demonstrated in Fig 1A–1C. All patients underwent chest CT scans, as detailed in Table 2. Multiple lobar lesions were observed in 73.0% of the infections. The most common

**Table 1. Demographics, clinical diagnosis, treatment, and outcome for patients with pulmonary mucormycosis.**

| Case | Age (y)/Sex | T1 (d) | T2 (d) | Immune status | Site of infection | Identification method | Treatment | Surgery | Outcome |
|---|---|---|---|---|---|---|---|---|---|
| 1 | 61/M | 1 | 18 | Suppression | Lung | Pathology | Posaconazole | No | Deceased |
| 2 | 35/F | 15 | 41 | Suppression | Disseminated | Pathology | Amphotericin B | Yes | Survivor |
| 3 | 31/F | 1 | 15 | Normal | Lung | NGS | Amphotericin B | No | Survivor |
| 4 | 44/M | 14 | 2 | Suppression | Lung | NGS | Amphotericin B | No | Survivor |
| 5 | 49/M | 20 | 3 | Normal | Lung | NGS | Isavuconazole | No | Survivor |
| 6 | 56/M | 3 | 28 | Suppression | Lung | NGS | Posaconazole | No | Deceased |
| 7 | 72/M | 20 | 6 | Normal | Lung | Culture | Amphotericin B | No | Survivor |
| 8 | 62/F | 15 | 21 | Normal | Lung | Pathology | Isavuconazole | No | Survivor |
| 9 | 22/F | 14 | 5 | Suppression | Lung | Pathology | Amphotericin B | No | Deceased |
| 10 | 74/M | 15 | 10 | Suppression | Lung | NGS | Amphotericin B | No | Deceased |
| 11 | 68/F | 30 | 6 | Suppression | Lung | Pathology | Amphotericin B, Posaconazole | No | Survivor |
| 12 | 38/F | 2 | 11 | Suppression | Lung | Pathology | Posaconazole, Amphotericin B | No | Survivor |
| 13 | 57/F | 20 | 3 | Suppression | Lung | NGS | Isavuconazole | No | Deceased |
| 14 | 73/M | 15 | 15 | Suppression | Lung | NGS | Amphotericin B | No | Deceased |
| 15 | 50/M | 8 | 9 | Normal | Lung | Pathology | Amphotericin B, Isavuconazole | No | Survivor |
| 16 | 26/M | 20 | 12 | Normal | Lung | Pathology | Amphotericin B, Isavuconazole | Yes | Survivor |
| 17 | 73/M | 90 | 20 | Normal | Lung | NGS | Isavuconazole | No | Survivor |
| 18 | 59/M | 20 | 4 | Suppression | Lung | NGS | Isavuconazole | No | Survivor |
| 19 | 59/M | 60 | 15 | Suppression | Lung | NGS | Isavuconazole | No | Survivor |
| 20 | 55/M | 20 | 7 | Normal | Lung | NGS | Isavuconazole | No | Survivor |
| 21 | 44/M | 4 | 7 | Suppression | Lung | Pathology | Isavuconazole | No | Survivor |
| 22 | 26/M | 2 | 19 | Normal | Lung | Pathology | Amphotericin B | Yes | Survivor |
| 23 | 80/F | 15 | 8 | Suppression | Lung | Culture | Isavuconazole | No | Deceased |
| 24 | 64/M | 7 | 12 | Normal | Lung | Culture | Isavuconazole | No | Survivor |
| 25 | 63/F | 7 | 15 | Suppression | Lung | Culture | Amphotericin B | No | Survivor |
| 26 | 56/M | 20 | 4 | Suppression | Lung | NGS | Isavuconazole | No | Survivor |
| 27 | 68/M | 21 | 9 | Suppression | Lung | Pathology | Amphotericin B, Isavuconazole | No | Survivor |
| 28 | 49/M | 2 | 27 | Suppression | Lung | NGS | Amphotericin B | No | Deceased |
| 29 | 56/M | 40 | 10 | Normal | Lung | NGS | Amphotericin B, Isavuconazole | No | Survivor |
| 30 | 70/F | 5 | 8 | Suppression | Lung | NGS | Amphotericin B | No | Survivor |
| 31 | 84/M | 10 | 13 | Normal | Lung | Culture | Amphotericin B | No | Survivor |
| 32 | 61/F | 10 | 14 | Normal | Lung | NGS | Isavuconazole, Amphotericin B | No | Deceased |
| 33 | 43/M | 3 | 25 | Suppression | Disseminated | NGS | Amphotericin B | No | Deceased |
| 34 | 65/M | 50 | 6 | Suppression | Lung | NGS | Isavuconazole | No | Survivor |
| 35 | 29/F | 21 | 12 | Normal | Lung | Culture | Isavuconazole | No | Survivor |
| 36 | 45/M | 2 | 15 | Normal | Lung | Pathology | Isavuconazole, Amphotericin B | Yes | Survivor |
| 37 | 75/M | 7 | 35 | Suppression | Lung | NGS | Amphotericin B, Posaconazole | No | Deceased |

Abbreviations: T1, time from symptom onset to hospital admission; T2, time from admission to diagnosis; F, female; M, male; NGS, next-generation sequencing

imaging findings included ground-glass infiltration (45.9%), nodules (48.6%), and patchy opacities (43.2%). Other notable imaging features were consolidation (40.5%), cavitary lesions (35.1%), reverse halo sign (40.5%), pleural effusion (32.4%), and lymphadenopathy (32.4%).

**Table 2. Clinical characteristics of patients with pulmonary mucormycosis and differences between 90-day survivors and non-survivors.**

| Patient demographics | Total (n = 37) | Patient survived (n = 26) | Patient deceased (n = 11) | P value |
|---|---|---|---|---|
| Male, n (%) | 25 (67.6) | 18 (69.2) | 7 (63.6) | 1 |
| Age, mean ± SD | 55.2 ± 16.2 | 53.5 ± 15.9 | 59.2 ± 16.9 | 0.336 |
| duration of hospitalization, mean ± SD | 30.3 ± 20.1 | 29.0 ± 19.0 | 33.5 ± 23.2 | 0.534 |
| clinical manifestations, n (%) | | | | |
| Fever | 26 (70.3) | 18 (69.2) | 8 (72.7) | 1 |
| Cough | 17 (45.9) | 14 (53.8) | 3 (27.3) | 0.138 |
| Hemoptysis | 11 (29.7) | 8 (30.8) | 3 (27.3) | 1 |
| Chest tightness | 13 (35.1) | 10 (38.5) | 3 (27.3) | 0.711 |
| Others | 9 (24.3) | 5 (19.2) | 4 (36.4) | 0.404 |
| Time from symptom onset to hospital admission, mean ± SD | 17.00 ± 18.03 | 20.15 ± 20.38 | 9.55 ± 6.64 | 0.102 |
| Time from admission to diagnosis, mean ± SD | 13.24 ± 8.97 | 11.62 ± 7.94 | 17.09 ± 10.45 | 0.090 |
| Underlying disease, n (%) | | | | |
| Diabetes | 14 (37.8) | 10 (38.5) | 4 (36.4) | 1 |
| Hypertension | 9 (24.3) | 4 (15.4) | 5 (45.5) | 0.091 |
| Hematological disorders | 11 (29.7) | 6 (23.1) | 5 (45.5) | 0.244 |
| Organ transplant | 7 (18.9) | 2 (7.7) | 5 (45.5) | 0.016 |
| Pulmonary diseases | 6 (16.2) | 6 (23.1) | 0 (0) | 0.151 |
| Tumor | 3 (8.1) | 2 (7.7) | 1 (9.1) | 1 |
| Others | 6 (16.2) | 4 (15.4) | 2 (18.2) | 1 |
| Identification method, n (%) | | | | 0.316 |
| NGS | 19 (51.4) | 11 (42.3) | 8 (72.7) | |
| Pathology | 12 (32.4) | 10 (38.5) | 2 (18.2) | |
| Culture | 6 (16.2) | 5 (19.2) | 1 (9.1) | |
| Laboratory test | | | | |
| WBC, mean ± SD | 8.49 ± 5.82 | 8.86 ± 5.56 | 7.63 ± 6.58 | 0.564 |
| NE, mean ± SD | 6.90 ± 5.46 | 7.17 ± 5.08 | 6.27 ± 6.50 | 0.653 |
| LY, mean ± SD | 0.93 ± 0.57 | 1.00 ± 0.52 | 0.75 ± 0.68 | 0.215 |
| EOS, mean ± SD | 0.04 ± 0.09 | 0.03 ± 0.06 | 0.07 ± 0.14 | 0.208 |
| HB, mean ± SD | 109.03 ± 28.43 | 114.54 ± 29.43 | 96.00 ± 21.86 | 0.069 |
| GM, mean ± SD | 2.00 ± 2.28 | 2.42 ± 2.57 | 1.02 ± 0.83 | 0.088 |
| CRP, mean ± SD | 66.92 ± 52.39 | 67.84 ± 54.41 | 64.74 ± 49.71 | 0.872 |
| PCT, mean ± SD | 0.42 ± 0.71 | 0.50 ± 0.84 | 0.24 ± 0.20 | 0.322 |
| ALB, mean ± SD | 32.35 ± 5.42 | 32.51 ± 4.84 | 31.97 ± 6.87 | 0.788 |
| Concurrent infections, n (%) | 30 (81.1) | 20 (76.9) | 10 (90.9) | 0.649 |
| Site of infection, n (%) | | | | 0.512 |
| Disseminated | 2 (5.4) | 1 (3.8) | 1 (9.1) | |
| Lung | 35 (94.6) | 25 (96.2) | 10 (90.9) | |
| Imaging findings, n (%) | | | | |
| Patchy opacities | 16 (43.2) | 10 (38.5) | 6 (54.5) | 0.475 |
| Ground-glass opacity (GGO) | 17 (45.9) | 13 (50) | 4 (36.4) | 0.447 |
| Cavity | 13 (35.1) | 8 (30.8) | 5 (45.5) | 0.465 |
| Reverse halo sign | 15 (40.5) | 10 (38.5) | 5 (45.5) | 0.728 |
| Flocculated opacities | 11 (29.7) | 8 (30.8) | 3 (27.3) | 1 |
| Nodule | 18 (48.6) | 14 (53.8) | 4 (36.4) | 0.331 |

*(Continued)*

| Patient demographics | Total (n = 37) | Patient survived (n = 26) | Patient deceased (n = 11) | P value |
|---|---|---|---|---|
| Crescent sign | 5 (13.5) | 3 (11.5) | 2 (18.2) | 0.623 |
| Vacuole | 8 (21.6) | 5 (19.2) | 3 (27.3) | 0.672 |
| Consolidation | 15 (40.5) | 10 (38.5) | 5 (45.5) | 0.728 |
| Pleural effusion | 12 (32.4) | 8 (30.8) | 4 (36.4) | 1 |
| Lymphadenopathy | 12 (32.4) | 8 (30.8) | 4 (36.4) | 1 |
| CT affected area, n (%) | | | | 0.688 |
| Single | 10 (27.0) | 8 (30.8) | 2 (18.2) | |
| Double | 27 (73.0) | 18 (69.2) | 9 (81.8) | |
| Treatment, n (%) | | | | |
| Amphotericin B | 22 (59.5) | 15 (57.7) | 7 (63.6) | 1 |
| Posaconazole | 5 (13.5) | 2 (7.7) | 3 (27.3) | 0.144 |
| Isavuconazole | 19 (51.4) | 16 (61.5) | 3 (27.3) | 0.057 |
| Suigery | 4 (10.8) | 4 (15.4) | 0 (0) | 0.296 |
| Immune status, n (%) | | | | 0.014 |
| Normal | 15 (40.5) | 14 (53.8) | 1 (9.1) | |
| Suppression | 22 (59.5) | 12 (46.2) | 10 (90.9) | |

Abbreviations: NGS: next-generation sequencing; WBC: white blood cells; NE: neutrophils; LY: lymphocyte; EOS: eosinophil; HB: hemoglobin; GM: galactomannan; CRP: C-reactive protein; PCT: procalcitonin; ALB: albumin.

## Histopathology and microbiology

The majority of patients underwent bronchoscopy. Fig 1D shows the bronchoscopic appearance of the airway mucosa in mucormycosis patients. Fig 1E depicts a nodular, neoplasm-like lesion along with necrotic material obstructing the right upper lobe bronchus. Bronchoscopy with bronchoalveolar lavage was performed in all cases, and the lavage fluid was sent for microbiological analysis. Nineteen of the 37 subjects were diagnosed using NGS. Among the nine patients confirmed with pulmonary mucormycosis, seven were diagnosed through bronchoscopic biopsy, and two were diagnosed via surgical biopsy of living tissue. Six patients were diagnosed via culture methods. The median time from admission to confirmed diagnosis was 13 days, ranging from 2 to 41 days. Depending on the sample type and laboratory conditions, we employed traditional culture methods, NGS, and histopathological examination for species identification of fungal genera. At the genus level, the identified species included Rhizopus (51.4%), Mucor (10.8%), and Rhizomucor (5.4%), along with other genera such as Rhizophthora (2.7%) and HDC fungi (2.7%). Additionally, eight isolates (21.6%) were classified as "unclassified Mucorales."

## Treatment and outcome

All patients received treatment, which included antifungal drugs, surgery, or a combination of both. Of the 37 patients, 13 were treated with amphotericin B lipid complex alone, 13 with isavuconazonium alone, 2 with posaconazole alone, 6 with a combination of amphotericin B lipid complex and isavuconazonium, 3 with a combination of amphotericin B lipid complex and posaconazole, and 4 underwent surgical treatment. In total, 11 patients (29.7%) died within 90 days of diagnosis. Among these, 9 patients succumbed to uncontrolled pulmonary mucormycosis, and 2 patients died due to massive hemoptysis. Fig 2 illustrates the progression of chest CT scans in a young male patient with diabetes who was diagnosed with pulmonary mucormycosis. The images show significant improvement following surgical intervention after initial antifungal therapy proved unsuccessful.

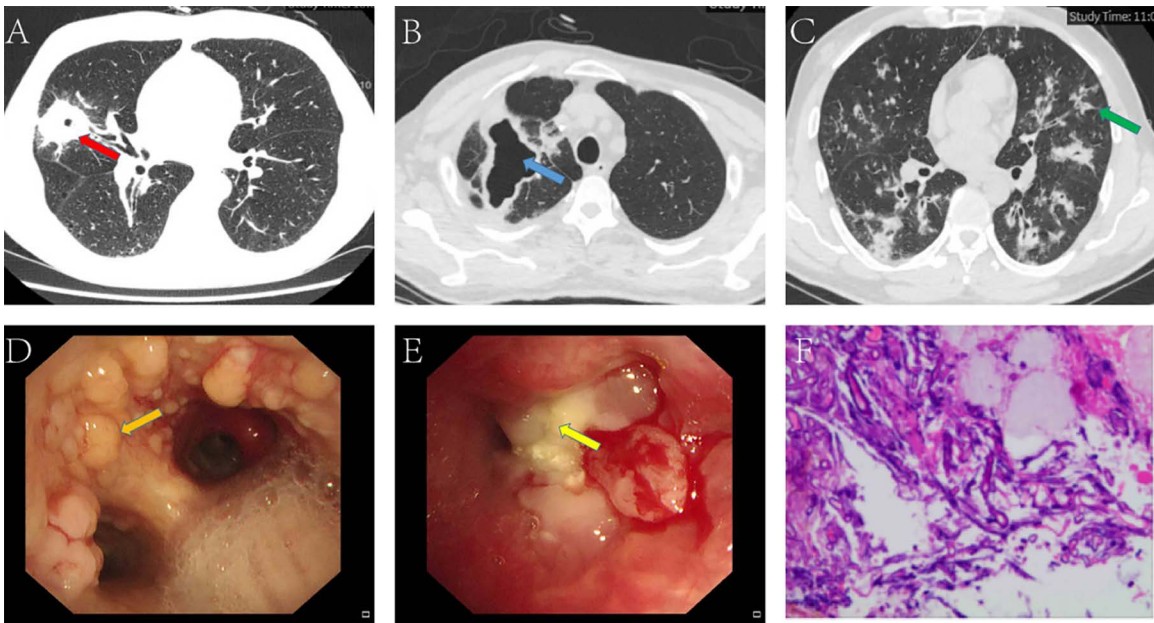

**Fig 1. CT Imaging and histopathology in patients with pulmonary mucormycosis. (A)** The red arrow indicates ground-glass opacity and a cavitary lesion centered on a mass in the upper lobe of the right lung, with surrounding consolidation and a reverse halo sign. **(B)** The blue arrow highlights the formation of a cavity within the lesion in the upper lobe of the right lung. **(C)** The green arrow points to a nodule exhibiting a halo sign. Bronchoscopic Images: **(D)** The orange arrow marks a nodular bulge. **(E)** The yellow arrow identifies the site of pathological sampling. Pathological Results: **(F)** Under PAS staining (400×magnification), thick hyphae with minimal or no septa and right-angled branching between the hyphae are observed.

## Predictive risk factors for outcome

In our study, 26 patients were discharged in stable condition, while 11 patients died. We compared the clinical data of survivors and non-survivors (Table 3). Univariate analysis of 90-day mortality identified several predictors of death: hypertension (HR = 3.36, 95% CI = 1.01–11.12, P = 0.048), organ transplantation (HR = 4.93, 95% CI = 1.48–16.4, P = 0.009), and immunosuppression (HR = 8.83, 95% CI = 1.13–69.14, P = 0.038). Immunosuppression significantly worsened the survival of pulmonary mucormycosis patients compared to those under immunocompetent conditions (Fig 3).

## Discussion

Mucormycosis is an invasive fungal disease often overlooked due to its rarity, yet its rapid progression is associated with high mortality rates [13]. Its incidence has been rising in recent years [14], particularly in 2020 following the COVID-19 pandemic, which saw an abnormal spike in cases [15,16]. The all-cause mortality rate of mucormycosis ranges between 40% and 80%, depending on underlying conditions and the site of infection [3]. Historically, pulmonary mucormycosis had a mortality rate of 56–76% [17,18], but recent trends indicate a decrease to 29–38% [19–21]. In this study, 29.7% (11/37) of pulmonary mucormycosis patients died within 90 days after diagnosis, consistent with recent reports.

This decline in mortality may be attributed to improved recognition and diagnostic techniques, facilitating early identification and treatment. Advances in antifungal therapy, enhanced patient support, better management of immunosuppression in organ transplant patients, and multidisciplinary collaboration have also contributed to better outcomes. However, mucormycosis remains a significant health concern, requiring continuous surveillance and research.

In our study, the median age of the patients was 55.2 years, aligning with previous reports [22]. The majority of patients were male (67.6%), consistent with other regional studies [6]. A recent murine study suggested female mice had a

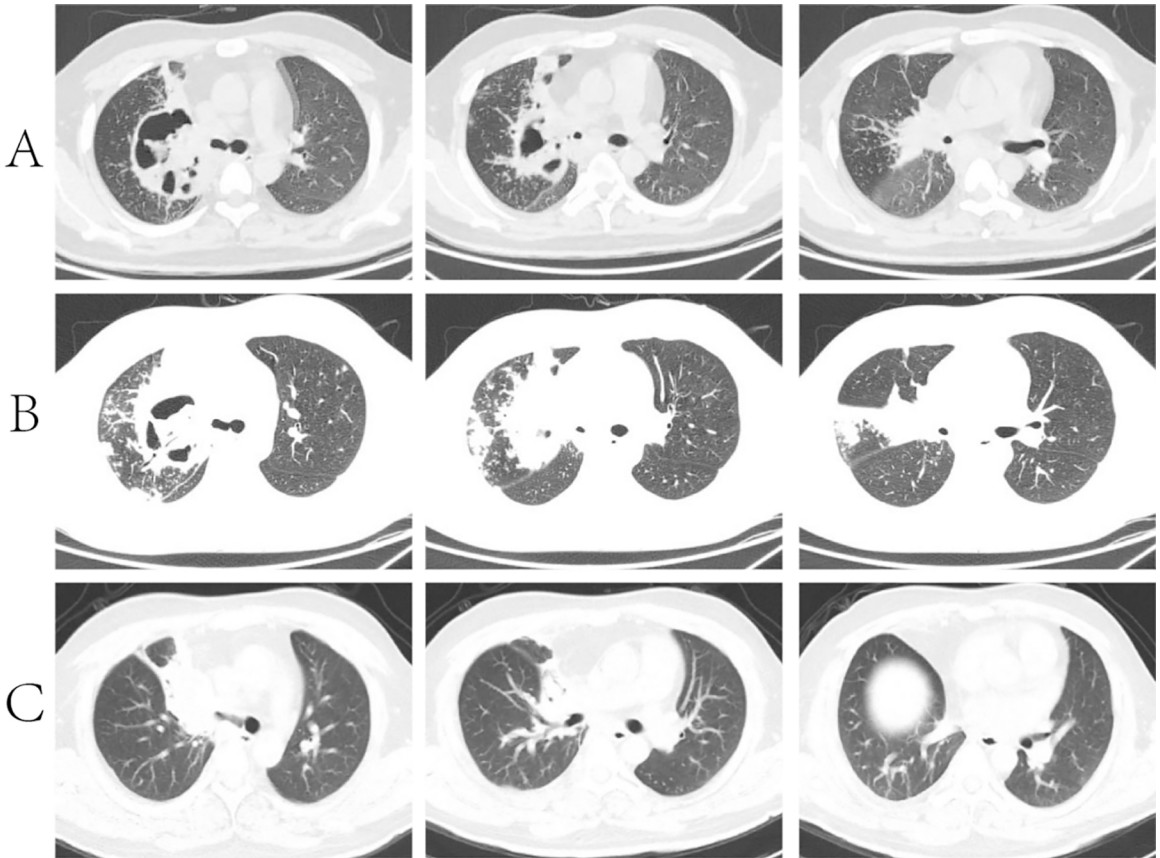

**Fig 2. Imaging of a 26-year-old male patient with diabetes and pulmonary mucormycosis. (A)** At admission, chest CT revealed a cavitary lesion in the upper lobe of the right lung, containing liquefied necrosis. **(B)** One week after admission, CT imaging showed an increase in the range of inflammatory lesions, with multiple small nodular lesions in the right lung. **(C)** One month after surgical treatment, the lesion was significantly reduced.

stronger anti-infective capacity and a longer median survival time than male mice in the context of diabetic ketoacidosis mucormycosis [23]. Whether estrogen has a protective effect on mucormycosis infection requires further exploration.

The most common clinical symptoms of pulmonary mucormycosis are cough, fever, and hemoptysis, which are similar to those of other molds, as noted in previous studies [6,24]. Diabetes was the most common underlying condition in our cohort (37.8%), followed by hematologic malignancies (29.7%), reflecting national trends [17,25]. Previous literature has identified diabetes mellitus as a potential adverse prognostic factor for pulmonary mucormycosis. However, our study did not demonstrate a significant correlation between diabetes and pulmonary mucormycosis outcomes. This discrepancy may be attributed to the lack of investigation into glycemic control status in our current research. Future studies are warranted to elucidate the relationship between glycemic control and pulmonary mucormycosis prognosis, with particular attention to the potential impact of long-term blood glucose management on patient outcomes.

Although risk factors for pulmonary mucormycosis patients are well established, different studies have reported varying mortality risk factors. One study found that poor outcomes were prevalent among patients with prolonged neutropenia and those receiving empirical voriconazole [26]. In a Pakistani cohort, mortality was associated with immunosuppression, thrombocytopenia, or the need for mechanical ventilation [27]. In our study, we identified hypertension, organ transplantation, and immunosuppression as predictors of 90-day mortality in pulmonary mucormycosis patients, with immunosuppression emerging as a significant risk factor for death. This finding aligns with previous research emphasizing the host's

**Table 3. Univariate Cox Regression analysis of 90-day mortality risk factors in patients with pulmonary mucormycosis.**

| Parameter | HR (95%CI) | *P* (Wald's test) |
|---|---|---|
| Sex, female | 1.31 (0.38,4.49) | 0.663 |
| Age | 1.02 (0.98,1.06) | 0.314 |
| Hemoptysis | 0.81 (0.22,3.06) | 0.757 |
| Diagnosis, confirmed diagnosis | 0.27 (0.03,2.13) | 0.215 |
| Time from hospital admission to diagnosis | 1.04 (0.98,1.09) | 0.195 |
| Diabetes | 0.94 (0.27,3.21) | 0.92 |
| Hypertension | 3.36 (1.01,11.12) | 0.048 |
| Hematological disorders | 2.07 (0.63,6.8) | 0.229 |
| Organ transplant | 4.93 (1.48,16.4) | 0.009 |
| WBC | 0.97 (0.86,1.09) | 0.601 |
| NE | 0.98 (0.86,1.11) | 0.724 |
| LY | 0.4 (0.12,1.35) | 0.139 |
| HB | 0.98 (0.96,1) | 0.087 |
| Concurrent infections | 2.63 (0.34,20.55) | 0.357 |
| Immune status, suppression | 8.83 (1.13,69.14) | 0.038 |
| CT affected area: double | 1.59 (0.34,7.38) | 0.552 |

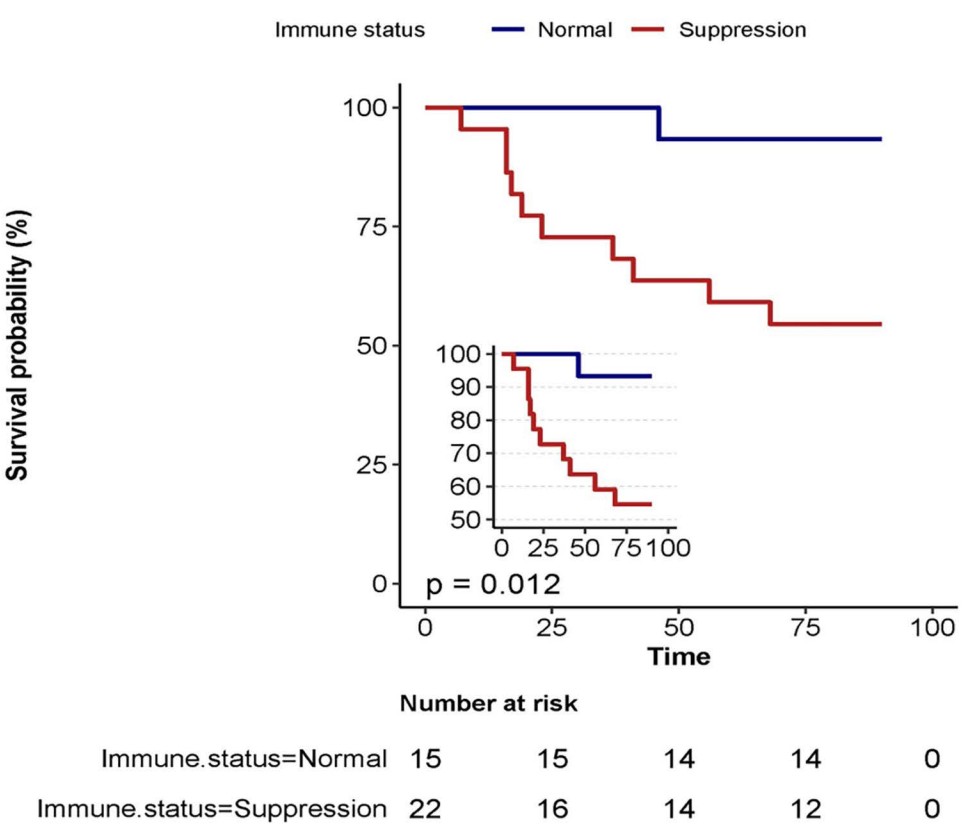

**Fig 3. Kaplan-Meier Survival Analysis Based on Immune Status.** The two groups represent patients stratified by their immune status at the time of admission. The survival curves illustrate differences in survival outcomes between these groups.

immune status as a key determinant of prognosis. However, in a study by Elzein et al., trauma was the primary risk factor, likely reflecting the specialized nature of the trauma center and the population studied, where most patients had normal immune function [28].

In the context of organ transplantation, the use of immunosuppressant drugs can weaken the body's natural defenses, making patients more susceptible to infections like mucormycosis. While these drugs are essential for preventing organ rejection, they also contribute to increased mortality in pulmonary mucormycosis patients. Although diabetes is commonly recognized as a risk factor for pulmonary mucormycosis [29], it was not identified as a mortality risk factor in our cohort. Interestingly, despite the increased incidence of hypertension in diabetic populations, our study highlights hypertension as an independent and significant risk factor for mortality in pulmonary mucormycosis patients. Hypertension is often associated with a chronic inflammatory state, which may impair the activity and function of immune cells, leading to immune dysregulation and reduced capacity to clear pathogens, thereby weakening the body's immune response. Additionally, patients with hypertension typically require long-term use of antihypertensive medications, some of which (e.g., beta-blockers or calcium channel blockers) may indirectly affect the immune system. However, the specific mechanisms underlying these effects require further investigation to validate and elucidate. Our study underscores the critical role of immune status in the prognosis of patients with pulmonary mucormycosis, emphasizing the importance of early intervention in high-risk patients.

In our cohort, only 16.2% of patients had positive microbial cultures, which is lower than the 78.78% reported by Abanamy et al. [30]. The low positive rate in our study may be due to the lack of molecular pathogen testing, as 51.4% of patients were diagnosed with pulmonary mucormycosis via NGS. NGS is more sensitive than traditional methods and significantly improves the diagnosis rate, though clinical settings must strengthen the interpretation of sequencing reports.

Mucormycosis imaging typically shows nodules, masses, consolidation, halo sign, reversed halo sign, and air crescent sign, all related to angioinvasion and tissue necrosis. These imaging changes are not specific to mucormycosis and overlay with those of invasive pulmonary aspergillosis, complicating the differentiation between the two. This is also a challenge in the imaging diagnosis of mucormycosis. Our study found that 81.1% of patients had coinfections with other pathogens, indicating the necessity for early identification and management of mixed infections.

Treatment options for mucormycosis remain very limited. Liposomal amphotericin B (LamB) combined with surgery is the recommended first-line treatment [31]. Only four patients underwent lobectomy in our study, so the benefits of combined antifungal and surgical could not be assessed. Multifocal lung involvement, present in 84.8% of our patients, limited surgical options, making early and effective antifungal treatment more crucial. Although combined antifungal and surgical debridement is common in other regions, such as in Saudi Arabia [30], the difference may be due to the higher prevalence of rhino-orbital-cerebral mucormycosis in those areas. Further research is needed to optimize the treatment strategies for different types of mucormycosis.

The study has limitations. It is retrospective, so not all information can be obtained from the medical records. We included patients who met the strict pulmonary mucormycosis definition and had detailed case information. Additionally, morphologically, mucormycosis cannot be completely distinguished from aspergillosis, and not all cases can be subjected to molecular testing. Additionally, This study carries a potential risk of misclassification within the genus Mucor. Indeed 8 cases were "unclassified" Mucorales but may also not be Mucorales at all, given the difficulties of anatomopathological identification. The limited sample size also reduces the statistical power of this analysis. Despite these limitations, future multicenter studies with more cases and advanced molecular testing will help identify host-related factors associated with poor outcomes.

Early diagnosis of mucormycosis is crucial, especially for immunosuppressed patients who are prone to secondary fungal infections with poor prognosis. When mucormycosis is suspected and tissue is available, NGS can be employed for accurate species identification, providing critical insights into the infection source. Early diagnosis, reversal of host factors, surgical resection of infected lesions, and timely antifungal treatment are key to improving outcomes. These four elements

are essential throughout the diagnosis and treatment process, and more efforts must be made to promote management to improve patient outcomes.

## Conclusion

Mucormycosis is a rare and often overlooked infection with a high mortality rate. Clinicians must enhance their recognition and management of mucormycosis, particularly in immunocompromised patients who are at higher risk for pulmonary mucormycosis. Definitive diagnosis relies on invasive histopathology and culture methods, while the use of molecular biology techniques in serum or tissue samples is emerging as an effective diagnostic tool. We recommend that when mucormycosis is suspected, appropriate samples should be collected promptly, and tests such as biopsy, culture, or NGS should be performed to confirm the diagnosis. Early diagnosis and timely administration of effective antimucormycotic therapy are vital for improving the prognosis of pulmonary mucormycosis patients.

## Acknowledgments

None.

## Author contributions

**Conceptualization:** Junjun Wan, Teng Liu, Fang Li.

**Data curation:** Junjun Wan, Teng Liu, Fang Li.

**Formal analysis:** Junjun Wan, Teng Liu, Fang Li, Shaohua Xu.

**Investigation:** Junjun Wan, Teng Liu, Fang Li.

**Validation:** Junjun Wan, Shaohua Xu.

**Writing – original draft:** Junjun Wan, Shaohua Xu.

**Writing – review & editing:** Teng Liu, Fang Li.

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
