## [Decision Letter · Decision Letter 0]

18 Mar 2025

PONE-D-24-59709Diagnosis, Clinical Features, and Mortality Risk Factors in a Chinese Cohort with Pulmonary MucormycosisPLOS ONE

Dear Dr. Xu,

Thank you for submitting your manuscript to PLOS ONE. After careful consideration, we feel that it has merit but does not fully meet PLOS ONE’s publication criteria as it currently stands. Therefore, we invite you to submit a revised version of the manuscript that addresses the points raised during the review process.

We look forward to receiving your revised manuscript.

Kind regards,

Felix Bongomin, MB ChB, MSc, MMed, FECMM

Academic Editor

PLOS ONE

3. In the online submission form, you indicated that [Our data are not publicly available due to patient privacy concerns and restrictions imposed by the ethics committee of our institution. Access to the data can be granted upon reasonable request and approval by the corresponding author, provided it complies with ethical guidelines and data protection regulations.].

Additional Editor Comments:

1. Please write pulmonary mucormycois in full, all through the manuscript.

2. Structure your abstract ... and describe variables in both frequency and %, also include all effect sizes , 95%CI and p-value.

3. Cite major papers eg Denning et al Lancet ID for burden of IFI including mucormycosis and the systematic review that inform WHO priority fungal pathogen list

Reviewers' comments:

Reviewer's Responses to Questions

**Comments to the Author**

1. Is the manuscript technically sound, and do the data support the conclusions?

Reviewer #1: Yes

2. Has the statistical analysis been performed appropriately and rigorously? 

Reviewer #1: Yes

3. Have the authors made all data underlying the findings in their manuscript fully available?

Reviewer #1: No

4. Is the manuscript presented in an intelligible fashion and written in standard English?

Reviewer #1: Yes

5. Review Comments to the Author

Reviewer #1: The manuscript is well organized and contributes valuable findings on pulmonary mucormycosis. It provides valuable insights into pulmonary mucormycosis. The manuscript is is systematically presented and well-written overall.

6. PLOS authors have the option to publish the peer review history of their article (what does this mean?). If published, this will include your full peer review and any attached files.

Reviewer #1: No

---

## [Author Response · Author response to Decision Letter 0]

26 Mar 2025

Esteemed Editors and Reviewers,

We extend our heartfelt gratitude for your thorough review and invaluable feedback on our manuscript titled "Diagnosis, Clinical Features, and Mortality Risk Factors in a Chinese Cohort with Pulmonary Mucormycosis" (Manuscript Number: PONE-D-24-59709). In accordance with the suggestions provided by the reviewers and editors, we have meticulously revised our manuscript. Herein, we address each of the reviewers' comments point by point. It is our sincere hope that these revisions meet the journal's standards and contribute to the enhancement of the manuscript's overall quality.

1.Reviewer's Responses to Questions and Comments to the Author

Questions 1: Is the manuscript technically sound, and do the data support the conclusions?

Reviewer #1: Yes

Author's reply: Thank you to the reviewers for their positive feedback on our research. We have ensured that all experimental designs and data analyses adhere to the rigorous standards of scientific research, and all conclusions are data-driven. In the revised manuscript, we have further clarified the details of the experimental methods and data analysis to ensure the reliability of our conclusions.

Questions 2: Has the statistical analysis been performed appropriately and rigorously?

Reviewer #1: Yes

Author's reply: We have re-examined all statistical analysis procedures and ensured their appropriateness and rigor. All statistical methods have been described in detail.

Questions 3: Have the authors made all data underlying the findings in their manuscript fully available?

Reviewer #1: No

Author's reply: Since the data involves patient privacy, we are unable to disclose the original data. However, we have updated the data availability statement in the revised manuscript, detailing the reasons for the data restrictions and providing avenues for data access. Specifically, the data can be made available upon reasonable request and with approval from the ethics committee. We have provided the contact information for the ethics committee as required by the journal.

Questions 4: Is the manuscript presented in an intelligible fashion and written in standard English?

Reviewer #1: Yes

Author's reply: We have conducted a comprehensive language polish on the manuscript to ensure it adheres to standard English writing conventions. All grammatical and spelling errors have been corrected, and the manuscript has been proofread by a professional English editor.

Questions 5: Review Comments to the Author

Reviewer #1: The manuscript is well organized and contributes valuable findings on pulmonary mucormycosis. It provides valuable insights into pulmonary mucormycosis. The manuscript is is systematically presented and well-written overall.

Author's reply: We extend our gratitude to the reviewers for their overall positive assessment of our manuscript. In accordance with the reviewers' suggestions, we have further refined the manuscript, particularly in the areas of data analysis and interpretation of results. We are confident that the revised manuscript is now more lucid and rigorous.

2. Additional Editor Comments

Comments 1: Please write pulmonary mucormycois in full, all through the manuscript.

Author's reply: We have ensured that "pulmonary mucormycosis" is written out in full throughout the manuscript, accompanied by its abbreviation in parentheses, to maintain terminological consistency.

Comments 2: Structure your abstract ... and describe variables in both frequency and %, also include all effect sizes , 95%CI and p-value.

Author's reply: We have structured the abstract according to the editor's suggestions and supplemented the frequencies, percentages, effect sizes, 95% confidence intervals, and p-values for all variables. The revised abstract is more comprehensive and clearer.

Comments 3: Cite major papers eg Denning et al Lancet ID for burden of IFI including mucormycosis and the systematic review that inform WHO priority fungal pathogen list.

Author's reply: We have supplemented our references with the work of Denning et al. on the burden of fungal infections and have cited the systematic review that informed the WHO's priority list of fungal pathogens. These citations further bolster the context and significance of our research.

3.Other revisions

We have adjusted the manuscript according to the journal's formatting requirements to ensure it aligns with the PLOS ONE format template. We have updated the data availability statement, detailing the reasons for data restrictions and providing access pathways to the data. We have reviewed and updated the reference list to ensure its completeness and accuracy. All cited references are up-to-date and have not been retracted.

We believe that these revisions have adequately addressed the comments from the reviewers and editors, significantly enhancing the quality of the manuscript. We look forward to your further feedback.

---

## [Editor Report · Decision Letter 1]

31 Mar 2025

PONE-D-24-59709R1Diagnosis, Clinical Features, and Mortality Risk Factors in a Chinese Cohort with Pulmonary MucormycosisPLOS ONE

Dear Dr. Xu,

Thank you for submitting your manuscript to PLOS ONE. After careful consideration, we feel that it has merit but does not fully meet PLOS ONE’s publication criteria as it currently stands. Therefore, we invite you to submit a revised version of the manuscript that addresses the points raised during the review process.

We look forward to receiving your revised manuscript.

Kind regards,

Felix Bongomin, MB ChB, MSc, MMed, FECMM

Academic Editor

PLOS ONE

Journal Requirements:

Additional Editor Comments:

DEAR AUTHORS

CHECK THE SPELLING OF MUCORMYCOSIS --THROUGH OUT THE MANUSCRIPT

DEFINE COVID-19 IN FULL IN FIRST USE

DEFINE HR AND CI IN THE ABSTRACT IN THE FIRST

---

## [Author Response · Author response to Decision Letter 1]

1 Apr 2025

Esteemed Editors and Reviewers,

We extend our heartfelt gratitude for your thorough review and invaluable feedback on our manuscript titled "Diagnosis, Clinical Features, and Mortality Risk Factors in a Chinese Cohort with Pulmonary Mucormycosis" (Manuscript Number: PONE-D-24-59709). In accordance with the suggestions provided by the reviewers and editors, we have meticulously revised our manuscript. Herein, we address each of the reviewers' comments point by point. It is our sincere hope that these revisions meet the journal's standards and contribute to the enhancement of the manuscript's overall quality.

1.Please review your reference list to ensure that it is complete and correct. If you have cited papers that have been retracted, please include the rationale for doing so in the manuscript text, or remove these references and replace them with relevant current references. Any changes to the reference list should be mentioned in the rebuttal letter that accompanies your revised manuscript. If you need to cite a retracted article, indicate the article’s retracted status in the References list and also include a citation and full reference for the retraction notice.

Re: We have reviewed the reference list to ensure its completeness and accuracy. All cited references are up-to-date and have not been retracted.

2.CHECK THE SPELLING OF MUCORMYCOSIS --THROUGH OUT THE MANUSCRIPT

Re: We have thoroughly checked the manuscript and corrected the spelling of "mucormycosis" throughout. It is now consistent and accurate.

3.DEFINE COVID-19 IN FULL IN FIRST USE

Re: We have defined "COVID-19" in full as "Coronavirus Disease 2019" at its first mention in the manuscript, as per your request.

4.DEFINE HR AND CI IN THE ABSTRACT IN THE FIRST

Re: We have defined "HR" (Hazard Ratio) and "CI" (Confidence Interval) in full at their first occurrence in the abstract.

We believe that these revisions have adequately addressed the comments from the reviewers and editors, significantly enhancing the quality of the manuscript. We look forward to your further feedback.

---

## [Editor Report · Decision Letter 2]

11 Apr 2025

Diagnosis, Clinical Features, and Mortality Risk Factors in a Chinese Cohort with Pulmonary Mucormycosis

PONE-D-24-59709R2

Dear Dr. Xu,

We’re pleased to inform you that your manuscript has been judged scientifically suitable for publication and will be formally accepted for publication once it meets all outstanding technical requirements.

Kind regards,

Felix Bongomin, MB ChB, MSc, MMed, FECMM

Academic Editor

PLOS ONE
---

## [Editor Report · Acceptance letter]

PONE-D-24-59709R2

PLOS ONE

Dear Dr. Xu,

I'm pleased to inform you that your manuscript has been deemed suitable for publication in PLOS ONE. Congratulations! Your manuscript is now being handed over to our production team.

Kind regards,

on behalf of

Dr. Felix Bongomin

Academic Editor

PLOS ONE